# Evaluating vector competence for Yellow fever in the Caribbean

Gaelle Gabiane[1,2], Chloé Bohers[1], Laurence Mousson[1], Thomas Obadia[3,4], Rhoel R. Dinglasan [5], Marie Vazeille [1], Catherine Dauga[1], Marine Viglietta[1], André Yébakima[6], Anubis Vega-Rúa[7], Gladys Gutiérrez Bugallo [7,8], Rosa Margarita Gélvez Ramírez [9,10], Fabrice Sonor[11,12], Manuel Etienne[11], Nathalie Duclovel-Pame[12], Alain Blateau[12], Juliette Smith-Ravin[13], Xavier De Lamballerie [10] & Anna-Bella Failloux [1] ✉

The mosquito-borne disease, Yellow fever (YF), has been largely controlled via mass delivery of an effective vaccine and mosquito control interventions. However, there are warning signs that YF is re-emerging in both Sub-Saharan Africa and South America. Imported from Africa in slave ships, YF was responsible for devastating outbreaks in the Caribbean. In Martinique, the last YF outbreak was reported in 1908 and the mosquito *Aedes aegypti* was incriminated as the main vector. We evaluated the vector competence of fifteen *Ae. aegypti* populations for five YFV genotypes (Bolivia, Ghana, Nigeria, Sudan, and Uganda). Here we show that mosquito populations from the Caribbean and the Americas were able to transmit the five YFV genotypes, with YFV strains for Uganda and Bolivia having higher transmission success. We also observed that *Ae. aegypti* populations from Martinique were more susceptible to YFV infection than other populations from neighboring Caribbean islands, as well as North and South America. Our vector competence data suggest that the threat of re-emergence of YF in Martinique and the subsequent spread to Caribbean nations and beyond is plausible.

Yellow fever virus (YFV), a positive-sense RNA virus of the genus *Flavivirus* (Family *Flaviviridae*) and its primary mosquito vector, *Aedes aegypti* (Linnaeus), were introduced to the Western Hemisphere from Africa via the slave trade[1]. The first documented outbreak of yellow fever (YF) occurred in the Yucatan Peninsula in 1648[2], followed by outbreaks in the United States and Canada between 1668 and 1870;[2] with mortality rates between 20% and 60%. In 1900, following the incrimination of *Ae. aegypti* as the vector for YFV[3,4], the Pan American Health Organization (PAHO) leveraged the insecticide DDT to achieve near total elimination of *Ae. aegypti* mosquito populations from the

[1]Institut Pasteur, Université Paris Cité, Arboviruses and Insect Vectors, Paris, France. [2]Université des Antilles, Ecole Doctorale 589, Schœlcher, Martinique, Marseille, France. [3]Institut Pasteur, Université Paris Cité, Bioinformatics and Biostatistics Hub, Marseille, France. [4]Institut Pasteur, Université Paris Cité, G5 Infectious Disease Epidemiology and Analytics, Paris, France. [5]University of Florida, Department of Infectious Diseases & Immunology and Emerging Pathogens Institute, College of Veterinary Medicine, Gainesville, FL, USA. [6]VECCOTRA, Rivière Salée, Martinique, Marseille, France. [7]Institut Pasteur de Guadeloupe, Laboratory of Vector Control Research, Unit Transmission Reservoir and Pathogens Diversity, Les Abymes, Guadeloupe, Marseille, France. [8]Department of Vector Control, Center for Research, Diagnostic, and Reference, Institute of Tropical Medicine Pedro Kouri, Havana, Cuba. [9]Centro de Atención y Diagnóstico de Enfermedades Infecciosas, Fundación INFOVIDA, Bucaramanga, Colombia. [10]Unité des Virus Emergents (UVE), Aix Marseille Université, IRD 190, Inserm 1207, IHU Méditerranée Infection, Marseille, France. [11]Centre de Démoustication et de Recherches Entomologiques, Lutte antivectorielle, Martinique, Marseille, France. [12]Agence Régionale de Santé, Direction de la Santé Publique, Martinique, Marseille, France. [13]Groupe de recherche Biospheres Université des Antilles, Campus de Schœlcher, Martinique, Marseille, France. ✉e-mail: anna-bella.failloux@pasteur.fr

Americas[5]. By 1957, most South American countries were free of *Ae. aegypti* except for residual foci in Argentina, Colombia, Venezuela, and Suriname[6]. However, the West Indies and the southeastern United States remained infested by *Ae. aegypti*. Armed with the 17D YF vaccine, which provides lifetime immunity[7], routine immunizations and vaccination campaigns across the globe have helped reduce the likelihood of YF outbreaks. Unfortunately, the cessation of mosquito control programs had allowed *Ae. aegypti* numbers to rebound and re-establish in urban areas[7].

In 2016, YF re-emerged in Sub-Saharan Africa with urban outbreaks in Angola and the Democratic Republic of Congo[8]. Similarly, in South America, YF outbreaks were reported in Brazil in 2016-2018 with more than 2000 confirmed cases, and a 25% death rate. However, in Brazil, transmission was mainly associated with the sylvatic cycle and not the urban cycle of YFV[9]. In 2018 alone, there were 109,000 severe infections and 51,000 deaths due to YF in both Africa and South America[10]. Declining vaccine coverage, unstable vaccine supplies, and an increase in imported cases in YF-free countries, which in turn, seeds local transmission, are a few of the key factors that have contributed to YF re-emergence[9].

Seven YFV genotypes have been identified in two geographic regions: five in Africa (West Africa I, West Africa II, East/Central Africa, Angola, East Africa) and two in the Americas (South America I and South America II)[11]. These strains show up to 21.5% nucleotide divergence and 5.2% differences in amino acid sequences[12]. Although the South American YFV strains are phylogenetically close to the West African strains as opposed to East African strains[11], it is unclear if all African YFV strains can be efficiently vectored by *Ae. aegypti* from the Americas. Recent studies have revealed underappreciated diversity among West African YFV isolates alone, wherein YFV strains have evolved differential growth kinetics in mammalian and insect cells as compared to canonically studied YFV strains[13]. This has important implications for the maintenance and spread of YFV. With >50% of initial YFV exposures resulting in mild or asymptomatic infections, molecular changes in the viral genome may induce additional alterations in YF pathogenesis leading to more severe outbreaks.

*Aedes aegypti* has two main subspecies that partition along morphological and ecological differences[14]. The cosmopolitan subspecies *Ae. aegypti aegypti* is well adapted to urban environments, breeds in small artificial containers, and is anthropophilic, whereas the African subspecies *Ae. aegypti formosus*, in forested habitats in sub-Saharan Africa, is zoophilic. The domestication of *Ae. aegypti aegypti* permitted its global expansion, first to the New World during the slave trade, then from East Africa to Asia, and its final expansion in the Pacific region during World War II[15]. Today, *Ae. aegypti* is the main vector of several arboviruses including dengue viruses (DENVs), chikungunya virus (CHIKV), Zika virus (ZIKV), and YFV[16]. Populations of *Ae. aegypti* have been firmly established throughout the Caribbean, resulting in periodic transmission and unexpected outbreaks of DENV, ZIKV, and CHIKV.

However, several studies have described variable competencies of geographically distinct *Ae. aegypti* populations for Zika[17] and DENV[12] in Africa, North, and South America. These variations in vector competence are thought be influenced in part by virus x vector host genotype interactions, including intrinsic mosquito tissue barriers (e.g., midgut and salivary glands) that prevent horizontal transmission of the virus in mosquito saliva[18].

Here, we investigated the variation of *Ae. aegypti* vector competence using fifteen mosquito populations (nine from Martinique, two from Haiti, one from Guadeloupe, one from Cuba, one from Florida (USA), and one from Colombia) that were established from field-collected mosquitoes (Table S1, Fig. 1). We exposed mosquitoes to an infectious blood meal containing one of the five YFV genotypes (Bolivia, Ghana, Nigeria, Sudan and Uganda; Fig. S1) and examined infection kinetics and dissemination rate at 14 and 21 days post-infectious feeding (Fig. S2). Infecting multiple mosquito populations with distinct YFV genotypes allowed us to evaluate which biological barrier (midgut or salivary glands) affects virus dissemination in the vector and subsequent presence in mosquito saliva, as a proxy of infectivity to next hosts.

## Results

### Aedes aegypti populations from Martinique were less susceptible to YFV Ghana

To measure vector competence, we detected the virus in three compartments leading to transmission: midguts (infection, IR), carcasses (dissemination, SDR) and saliva (transmission, STR). At 14 dpi, we showed that from the 45 population-virus pairings (9 populations x 5 YFV): i) all pairings had mosquitoes with infected midguts (IRs ranging from 3.0% to 100%; Fig. S4, Table S2), ii) 42 pairings produced

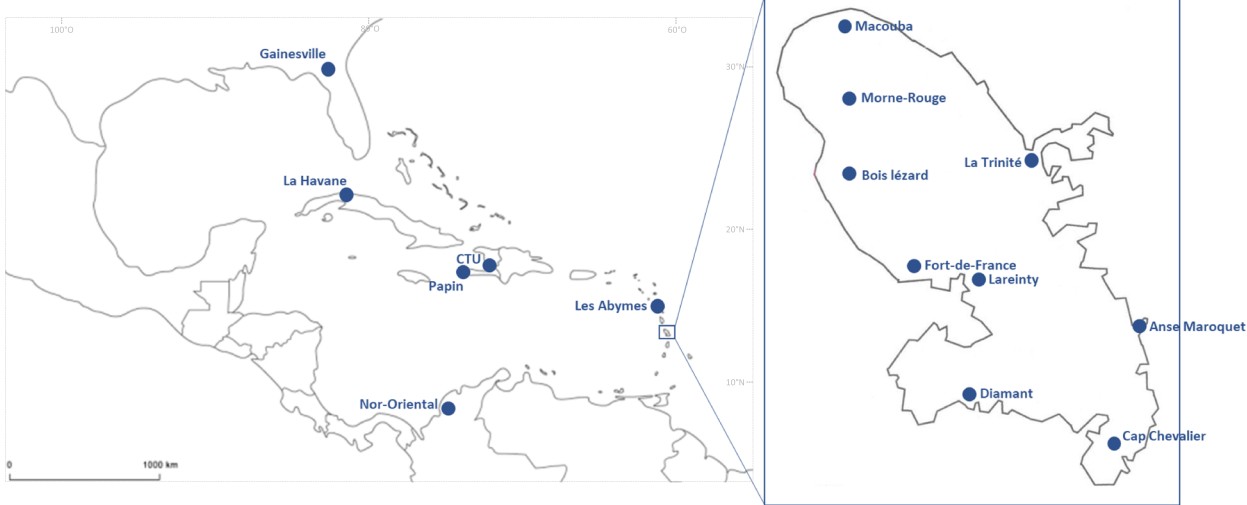

**Fig. 1 | Map showing the location of *Aedes aegypti* populations collected in Martinique in 2019-2020.** The nine populations were experimentally infected with five YFV genotypes. Maps were built using the open source map site: http://www. cartesfrance.fr/carte-france-departement/carte-departement-Martinique.html and https://d-maps.com/carte.php?num_car=15535&lang=fr.

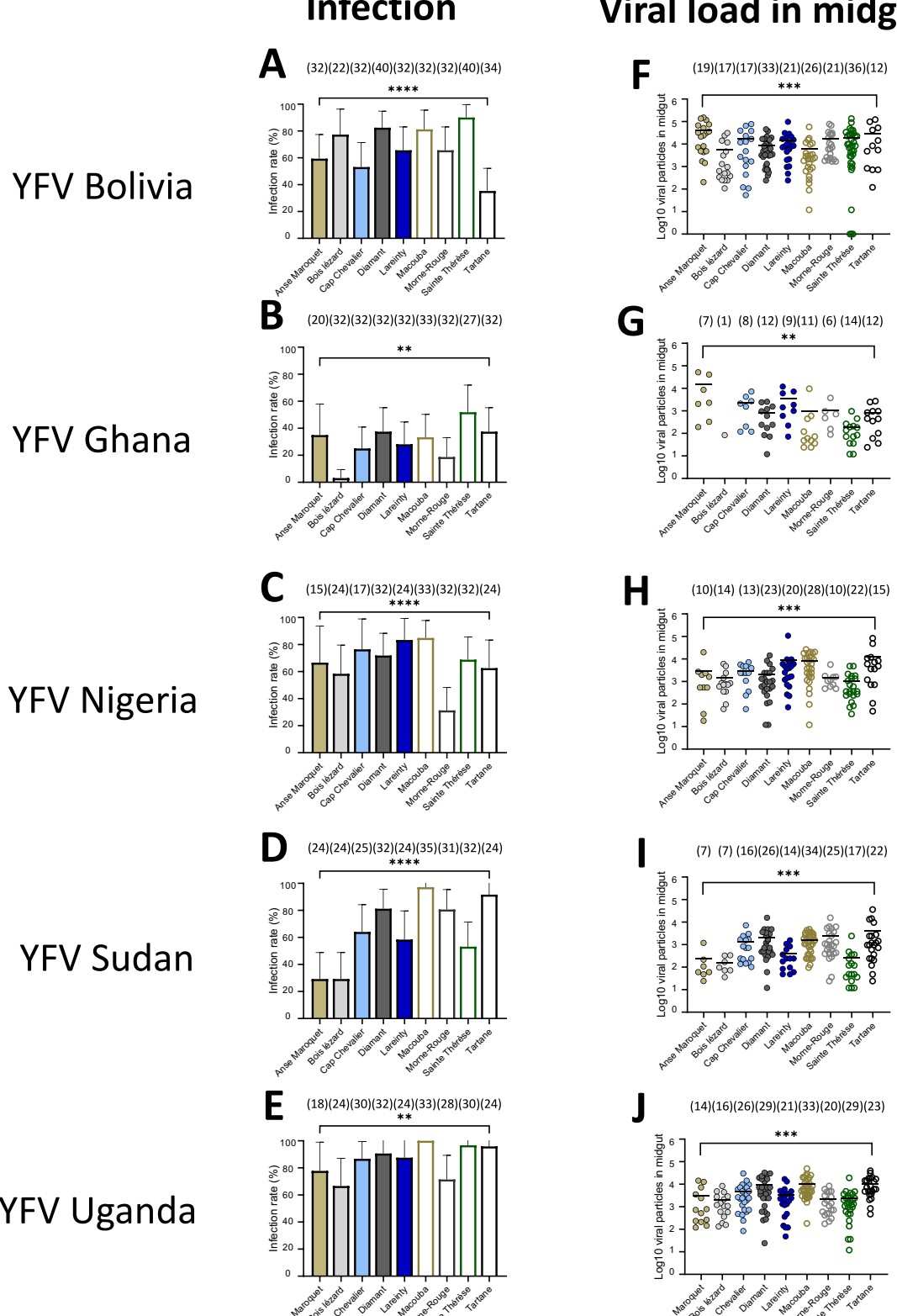

mosquitoes with infected carcasses (SDRs ranging from 16.7% to 100%; Fig. S5, Table S2), and *iii)* only 23 pairings had mosquitoes with infectious saliva (STRs ranging from 5.6% to 50%; Fig. S6, Table S2). We also examined the mean number of viral particles: *i)* in midguts which varied from 18 to 43,086 (Fig. S4, Table S2), *ii)* in the carcasses which varied from 20 to 45,378 (Fig. S5, Table S2), and *iii)* in saliva which varied from 2 to 376 (Fig. S6, Table S2).

By extending the examination of mosquitoes to 21 dpi, we observed that from the 45 population-virus pairings: *i)* all pairings produced mosquitoes with infected midguts (IRs ranging from 3.1% to 100%; Fig. 2, Table S3), *ii)* 43 pairings had mosquitoes with infected carcasses (SDRs ranging from 8.3% to 100%; Fig. 3, Table S3), and *iii)* 37 pairings had mosquitoes with infectious saliva (STRs ranging from 4.2% to 100%; Fig. 4, Table S3). We measured the mean number of

**Fig. 2 | Infection rates and viral loads in mosquito midguts of nine *Aedes aegypti* populations from Martinique examined 21 days after exposure to an infectious blood meal containing one YFV strain (Bolivia, Ghana, Nigeria, Sudan or Uganda).** Mosquitoes were exposed to an infectious blood meal at a titer of $10^7$ FFU/mL using an Hemotek system maintained at 37 °C. Engorged mosquitoes were kept for 21 days in controlled conditions and then dissected to isolate the midgut for estimating the viral load by titration. **A–E** The infection rate was defined as the proportion of mosquitoes with an infected midgut among the total number of mosquitoes exposed to the blood meal. The error bars correspond to the confidence intervals (95%) for IR (Table S3); **$0.001 \leq p < 0.01$ ($p = 0.005$ for B, $p = 0.001$ for E), ****$p < 0.0001$ (**A, C, D**) by Fisher's exact test (two-sided). **F–J** The number of viral particles in individual mosquito midguts (scatter plot) and mean (bar) are shown (Table S3); **$0.001 \leq p < 0.01$ ($p = 0.0011$ for G), ***$0.0001 \leq p < 0.001$ ($p = 0.0002$ for F and $p = 0.0001$ for **H–J**) by Kruskall-Wallis non-parametric test (one-sided). In brackets are the numbers of mosquitoes tested. Source data are provided as a Source Data file.

viral particles: *i)* in midguts which varied from 84 to 40,745 (Fig. 2, Table S3), *ii)* in carcasses which ranged from 32 to 199,921 (Fig. 3, Table S3), and *iii)* in saliva which varied from 2 to 1,626 (Fig. 4, Table S3).

Altogether, all five YFV strains infected and disseminated in the nine distinct mosquito populations. While four YFV were transmitted by the nine populations at 21 dpi with the highest IRs, SDRs, and STRs values for YFV Uganda and YFV Bolivia, YFV Ghana was only transmitted by two populations from Martinique (Diamant and Lareinty). In general, STR values were much lower than SDRs, and SDRs were similar to IRs (Table S3), indicating that the salivary glands played a more significant role as a barrier than the midgut to the virus dissemination in mosquitoes.

### YFV load in midgut determines efficient dissemination and transmission

When considering the number of viral particles in each compartment (midgut, carcass, and saliva) of *Ae. aegypti* populations from Martinique at 14 dpi, the mean viral load in the midgut was significantly higher in mosquitoes that could disseminate the virus, as compared to mosquitoes unable to disseminate the virus (Fig. S7A–E, Table S2), and in a similar way, in mosquitoes able to transmit as compared to mosquitoes unable to transmit the virus (Fig. S7F–J, Table S2). Similarly, at 21 dpi, the mean viral load in midgut was significantly higher in mosquitoes that could disseminate the virus (Fig. 5A–E, Table S3) and in mosquitoes that could transmit the virus in saliva (Fig. 5F–J, Table S3). In general, the mean number of virus particles in the midgut did not increase significantly from 14 to 21 dpi (Fig. S7). When examining the viral load in carcasses of mosquitoes that could transmit vs. mosquitoes that are unable to transmit at 14 dpi (Fig. S8A–E, Table S2) and 21 dpi (Fig. S8F–J, Table S3), all YFV except YFV Ghana required higher viral loads in the midgut to allow viral transmission (Fig. S8).

Logistic regression revealed the viral load in midguts to be an important predictor in explaining dissemination, and likewise the viral load in the carcass in explaining transmission: likelihood-ratio tests without the log10-transformed viral load from predictors resulted in a significantly worse deviance criterion ($P < 0.0001$ for both models). Collectively, dissemination and transmission of YFV required higher viral load in midguts. Owing to complex relationships between YFV genotypes and mosquito populations, the viral load required to achieve a model-predicted 50% rate of dissemination ranged ~250–8,000 particles. For transmission, higher values were required (>10,000 particles). Viral loads did not change when we extended the examination from 14 dpi to 21 dpi.

### Aedes aegypti populations from neighboring Caribbean islands and countries in the Americas were less efficient in transmitting YFV

To understand the vector competence of the nine *Ae. aegypti* populations from Martinique in a regional context, six other populations (Abymes, CTU, Gainesville, Havana, Nor Oriental, and Papin), were examined at 21 dpi. We showed that from the 30 population-virus pairings (6 populations x 5 YFV): *i)* all pairings produced mosquitoes with infected midguts (IRs ranging from 11.5% to 100%; Fig. 6A–E, Table S4), *ii)* all pairings had mosquitoes with infected carcasses (SDRs ranging from 6.7% to 100%; Fig. 6F–J, Table S4), and *iii)* 20 pairings had mosquitoes with infectious saliva (STRs ranging from 5.9% to 35.7% without any significant differences (Fig. 6K–O, Table S4). The mean number of viral particles varied: *i)* in midguts from 72 to 18,297 (Fig. 7A–E, Table S4), *ii)* in carcasses from 36 to 119,708 (Fig. 7F–J, Table S4), and *iii)* in saliva from 2 to 3,808 (Fig. 7K–O, Table S4). Taken together, we observed that YFV Bolivia and YFV Uganda infected, disseminated, and were successfully transmitted by all 6 populations (Fig. 6, Table S4); having generated the highest viral loads in the three compartments (Fig. 7, Table S4). YFV Ghana was only transmitted by the Papin population (Fig. 6L, Fig. 7L). Posterior probabilities for CTRs agreed with the observed data (Fig. S9). YFV Ghana transmission was null for all populations, with 95%-credibility interval [CrI] and hardly departing from that value for the Papin population (0.0%, 95%-CrI [0.0%–2.9%]). Mosquito populations from Martinique were overall more efficient at transmitting YFV Nigeria than others (with the exception for the Havana population), as highlighted by higher upper boundaries of 95%-CrI. Albeit with very low median TR, the corresponding CrIs for YFV Bolivia were consistently non-null, suggesting low heterogeneity in susceptibility to that strain at a regional scale. Highest CTRs were predicted for YFV Uganda across all mosquito populations, with particularly high CTRs in those sampled from Martinique where 6 populations had their median CTRs exceeding 25%. No other populations reached that value (Fig. S9).

## Discussion

Yellow fever is endemic today in 44 countries in Sub-Saharan Africa and South America where the vector *Ae. aegypti* is present, representing approximately 900 million people at risk of infection. In the Caribbean, Martinique survived its last YF outbreak in 1908. We showed that *Ae. aegypti* mosquitoes from Martinique were able to transmit five YFV genotypes (4 African and 1 American), albeit with a much lower transmission for YFV Ghana. We also showed that Martinique *Ae. aegypti* mosquitoes were more susceptible to YFV than the *Ae. aegypti* populations from neighboring Caribbean islands and countries in the Americas.

Despite past major successes in eliminating a mosquito-borne viral disease and an effective vaccine, YF epidemics continue to occur more than a century later. The last urban epidemics in the Americas and West Africa date back to 1928 (Brazil) and 2015 (West Africa)[9]. In Central Africa, outbreaks were reported in 2016-2019 in Angola[19] and the Democratic Republic of Congo (884 confirmed cases and 381 deaths) with human cases exported to China[20]. YF also affected neighboring countries such as Uganda[21] and Nigeria[22] in East and West Africa, respectively. Similarly, concurrent outbreaks were reported in the Americas, wherein humans were mainly infected by the bite of forest canopy-dwelling mosquitoes of the genera *Haemagogus* (primary vectors) and *Sabethes* (secondary vectors); both highly competent for American and African YFV strains[23]. In Brazil, the YFV genotype I outbreak started in the northern states of Brazil and spread to southern Brazil[24] causing 2,237 confirmed cases and 759 deaths[25]. In the Caribbean, the first epidemics began in 1640 in Guadeloupe, then Jamaica in 1655 and Martinique in 1687 and today, urban YF is no longer reported but the mosquito vector, *Ae. aegypti*,

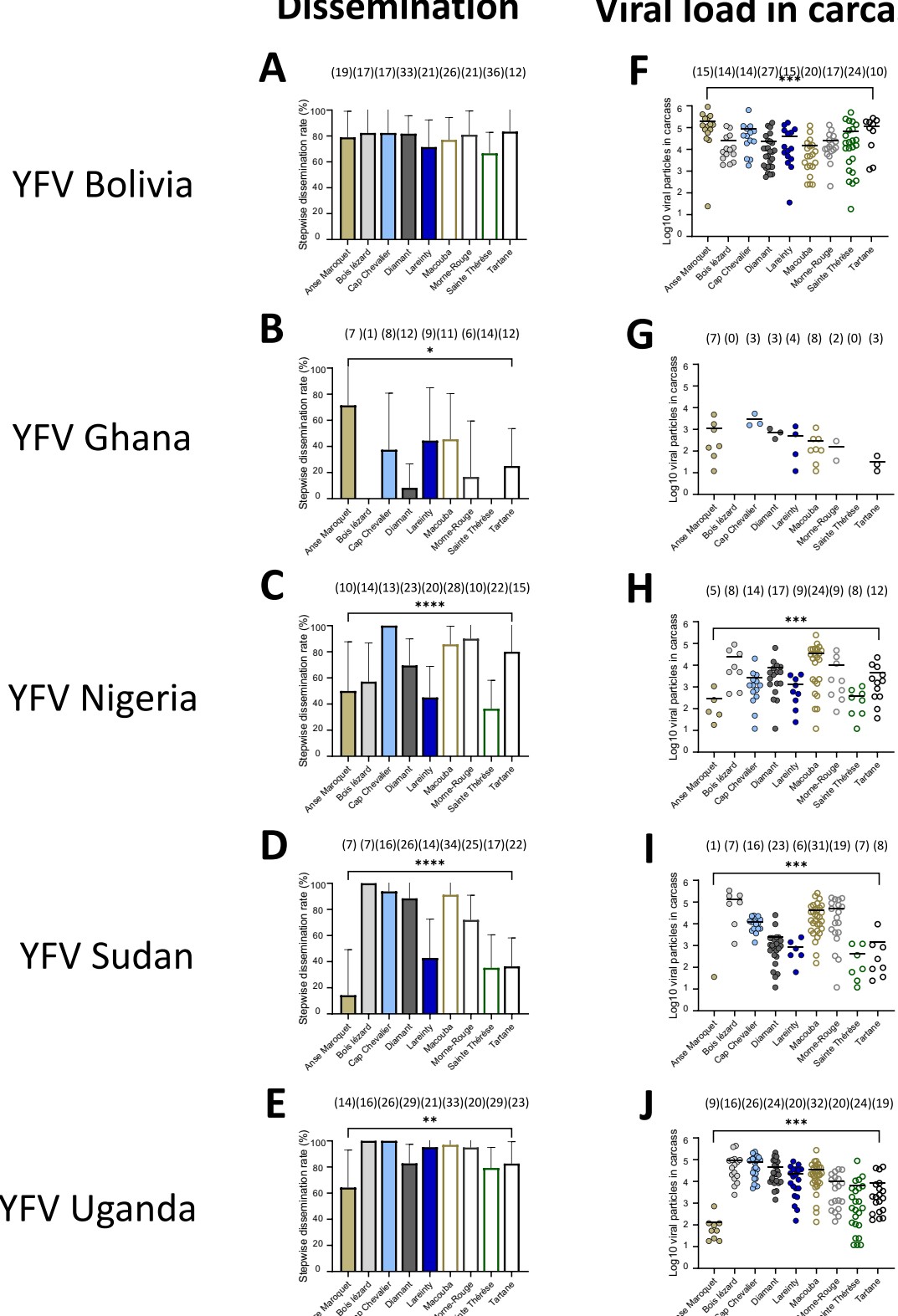

remains established. In Trinidad & Tobago, YF was primarily an urban disease before 1828. Since 1838, YFV (American II genotype) circulates between Alouatta monkeys and *Haemagogus* mosquitoes in a persistent sylvatic cycle in the country, which led to a large outbreak in 1979[26]. Between 1978-1980, following a YFV outbreak in Trinidad & Tobago, YFV reached the South American continent via northern Brazil (1989) and Venezuela (1992)[27]. The recent outbreaks in the Americas and Africa raised fears of urban outbreaks in the two continents, and the potential establishment of YF in Asia, where local YF has never been reported[28]. There are multiple factors that have led to YF re-emergence globally, but insufficient vaccine coverage is by far the most important one.

The French overseas department of French Guiana where vaccination is mandatory, is at risk of a greater number of YF imported

**Fig. 3 | Stepwise dissemination rates and viral loads in mosquito carcasses of nine *Aedes aegypti* populations from Martinique examined 21 days after exposure to an infectious blood meal containing one YFV strain (Bolivia, Ghana, Nigeria, Sudan or Uganda).** Mosquitoes were exposed to an infectious blood meal at a titer of $10^7$ FFU/mL using an Hemotek system maintained at 37 °C. Engorged mosquitoes were kept for 21 days in controlled conditions and then dissected to isolate the carcass for estimating the viral load by titration. **A–E** The stepwise dissemination rate was defined as the proportion of mosquitoes with an infected carcass among mosquitoes with an infected midgut. The error bars correspond to the confidence intervals (95%) for SDR (Table S3); *$0.01 \leq p < 0.05$ ($p = 0.018$ for **B**), **$0.001 \leq p < 0.01$ ($p = 0.004$ for **E**), ****$p < 0.0001$ (**C**, **D**) for by Fisher's exact test (two-sided). **F–J** The number of viral particles in individual carcasses (scatter plot) and mean (bar) are shown (Table S3); ***$0.0001 \leq p < 0.001$ ($p = 0.0001$ for **F**, $p = 0.0001$ for **H–J**) by Kruskall-Wallis non-parametric test (one-sided). In brackets are the numbers of mosquitoes tested. Source data are provided as a Source Data file.

cases, resulting in local cases, as was in 1998, the first YF outbreak in the country since 1902[29,30]. More than a few hundred passengers on more than 2–3 flights per day travel between French overseas department and French Guiana and nearby countries in South America. As such, the recent YF situation in French Guiana is worrisome for the other French overseas departments in the Caribbean. We showed that the nine *Ae. aegypti* populations from Martinique were able to transmit five YFV genotypes except for the YFV Ghana strain. YFV Ghana corresponds to the Asibi strain isolated from a patient with a mild infection in Ghana in 1927[31], which was then used to produce the 17D YFV vaccine after serial passages on embryonated chicken eggs[32]. The vaccine 17D differs from the Asibi strain by 75 mutations, and this strain may have retained the intrinsic property of poor infectiousness to mosquitoes[33]. Seven non-synonymous mutations are present in domain III of the envelope protein, which likely affects virus internalization by *Ae. aegypti* midgut cells. Apart from YFV Ghana, the four other YFV genotypes are effectively transmitted by *Ae. aegypti* Martinique populations, raising concern that imported human cases from countries in Latin America and sub-Saharan Africa can successfully infect Martinique vectors to initiate local transmission, since the highest infection efficiencies were obtained for YFV Uganda (East Africa genotype) and YFV Bolivia (America II genotype).

Bayesian modeling is well suited for nested processes and especially useful to propagate uncertainty originating from different processes into an outcome of interest. Here, owing to many interactions between YFV genotypes and mosquito populations (as highlighted by the large number of significant interaction terms in frequentist logistic regression), as well as the range of values to be investigated for viral loads in mosquito midguts and carcasses, such a framework was ideal to model the successive (conditional) probabilities of infection, dissemination, and transmission. An implicit limitation of our model resides in the representativity of viral loads (the only continuous predictors) used for model fitting: our model makes the underlying assumption that the variability of such titers across YFV genotypes and mosquito heterogeneity is well captured from experimental conditions, so that posterior distributions can be used when sampling outcomes of interest. With enough mosquitoes dissected, we expect these viral loads to be representative of their real-life counterpart. The complete experimental design at day 21 post-infectious feeding (and near-complete at day 14), with ~30 mosquitoes for each combination, combined with a simple set of conditional models, therefore allowed us to estimate not only the STRs, but also the intermediate outcomes (infection and dissemination). While not all pairs of mosquito populations and YFV genotypes were available at day 14 post-infection, the available results suggest that transmission at day 14 and at day 21 post-infectious feeding are largely comparable, with infected mosquitoes therefore being competent for transmitting YFV into human hosts well within their average lifetime.

The virus population undergoes bottlenecks and evolutionary pressures during replication cycles from the midgut to the salivary glands from where the virus is transmitted to the next host during blood feeding[34]. While a minimum level of mammalian host viremia is necessary to infect vectors[35], a minimum viral load in the vector is also required for efficient viral dissemination and horizontal transmission to a mammalian host[36]. We observed that a threshold of ~250–8,000

viral particles in the midgut permits viral dissemination and >10,000 viral particles is needed for viral transmission. If the mosquito ingests 2–3 μL of human blood[37] at a viremia around $10^6$–$10^8$ copies/mL[38], mosquito may absorb $10^3$–$10^5$ viral particles, which is in the range of the threshold determined from our study. Besides the viral load, the viral genotype can also play a critical role. We found that YFV Uganda is the most efficiently transmitted genotype overall, with the Nigerian and Sudanese genotypes being equally efficient in mosquitoes from Martinique. In contrast, *Ae. aegypti* mosquitoes from surrounding countries are less able to transmit these three African YFV genotypes. The regionally relevant YFV Bolivia strain, although not reaching high CTRs, is notably able to infect and disseminate into all studied mosquito populations, suggesting it is particularly adapted to *Ae. aegypti* from the Caribbean. Vector genetic factors involved in the success of viral transmission result from a long-term virus-vector interactions. Coevolution of vectors and the pathogens they transmit can positively modulate specific gene expression to maintain the vector fitness and secure pathogen transmission. While it is not of public health concern yet, due to its low ability to disseminate, YFV should remain under surveillance to promptly detect virus adaptation into local vector populations. Finally, we hypothesize that YFV Ghana strains will not pose a threat at this stage as it was largely unable to infect *Ae. aegypti* in our experiments. We also showed that the salivary glands of *Ae. aegypti* Martinique were clearly more efficient in limiting YFV transmission than the midgut was in limiting virus dissemination as compared to *Ae. aegypti* from Brazil[23], Guadeloupe[39], the Asia-Pacific region[40], and Central Africa[41].

It is well established that *Ae. aegypti* moved from Africa into the New World during the slave trade[15]. This mosquito implicated in urban outbreaks[42] was subjected to intensive vector control in the 1940s-50s leading to its elimination from many countries in the Americas and the Caribbean post 1960. Recolonization started in the 1970s after relaxation of vector control measures. *Ae. aegypti* from North America, South America, and the Caribbean are genetically structured as compared populations from the African native cradle[43]. We showed that *Ae. aegypti* from continental America, i.e., Colombia and the United States of America, and Caribbean countries such as Guadeloupe, Haiti, and Cuba were less susceptible to all five YFV genotypes, highlighting that the outcome of infection depends on the specific pairing of vector and virus genotypes[18]. In Martinique, *Ae. aegypti* populations are expected to derive from mosquitoes from North America and South America. These populations were shown to be genetically differentiated most likely as a result of recurrent insecticide treatments in response to dengue outbreaks[44]. Therefore, the effectiveness of the vector's intrinsic defense mechanism depends on the virus genotype and its infection dynamics. Knowledge of vector competence for local *Ae. aegypti* populations can help inform the use of limited resources for targeted vector control, which in the case of the Caribbean countries tested, appear most critical for Martinique.

However, *Ae. aegypti* populations have developed resistance to commonly used insecticides, and only a few chemical active ingredients are available to combat *Aedes*-borne diseases. In Martinique, resistance to organophosphates and pyrethroids has been reported since the 1980s and 1990s, respectively;[45] with both metabolic and target site-based resistance mechanisms implicated. *Bacillus*

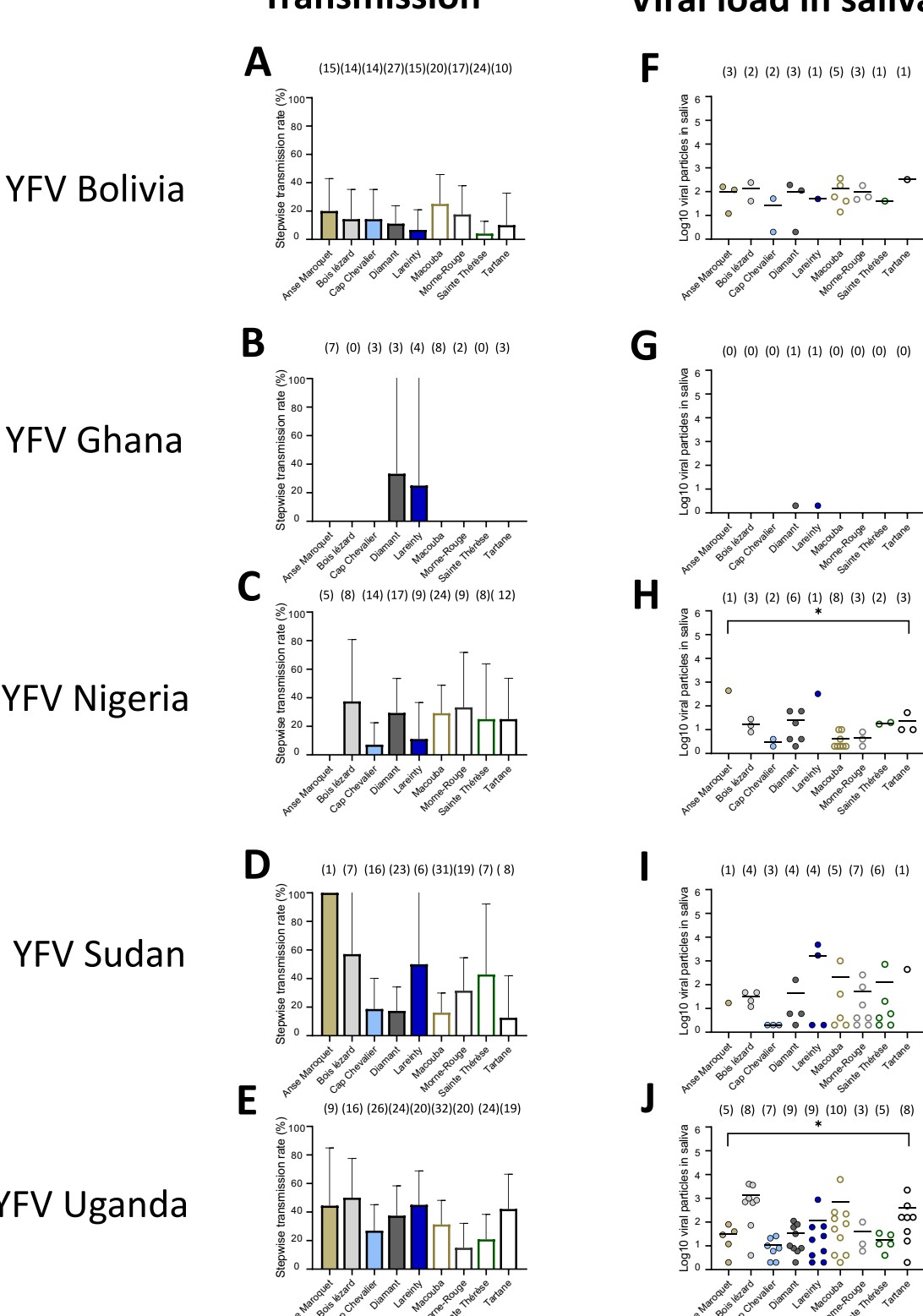

**Fig. 4 | Stepwise transmission rates and viral loads in mosquito saliva of nine *Aedes aegypti* populations from Martinique examined 21 days after exposure to an infectious blood meal containing one YFV strain (Bolivia, Ghana, Nigeria, Sudan or Uganda).** Mosquitoes were exposed to an infectious blood meal at a titer of $10^7$ FFU/mL using an Hemotek system maintained at 37 °C. Engorged mosquitoes were kept for 21 days in controlled conditions and then prepared for salivation to estimate the viral load by titration. **A**–**E** The stepwise transmission rate was defined as the proportion of mosquitoes with infectious saliva among mosquitoes with an infected carcass. The error bars correspond to the confidence intervals (95%) for STR (Table S3). **F**–**J** The number of viral particles in individual saliva (scatter plot) and mean (bar) are shown (Table S3); *$0.01 \leq p < 0.05$ ($p = 0.048$ for **H**) by Kruskall-Wallis non-parametric test (one-sided). In brackets are the numbers of mosquitoes tested. Source data are provided as a Source Data file.

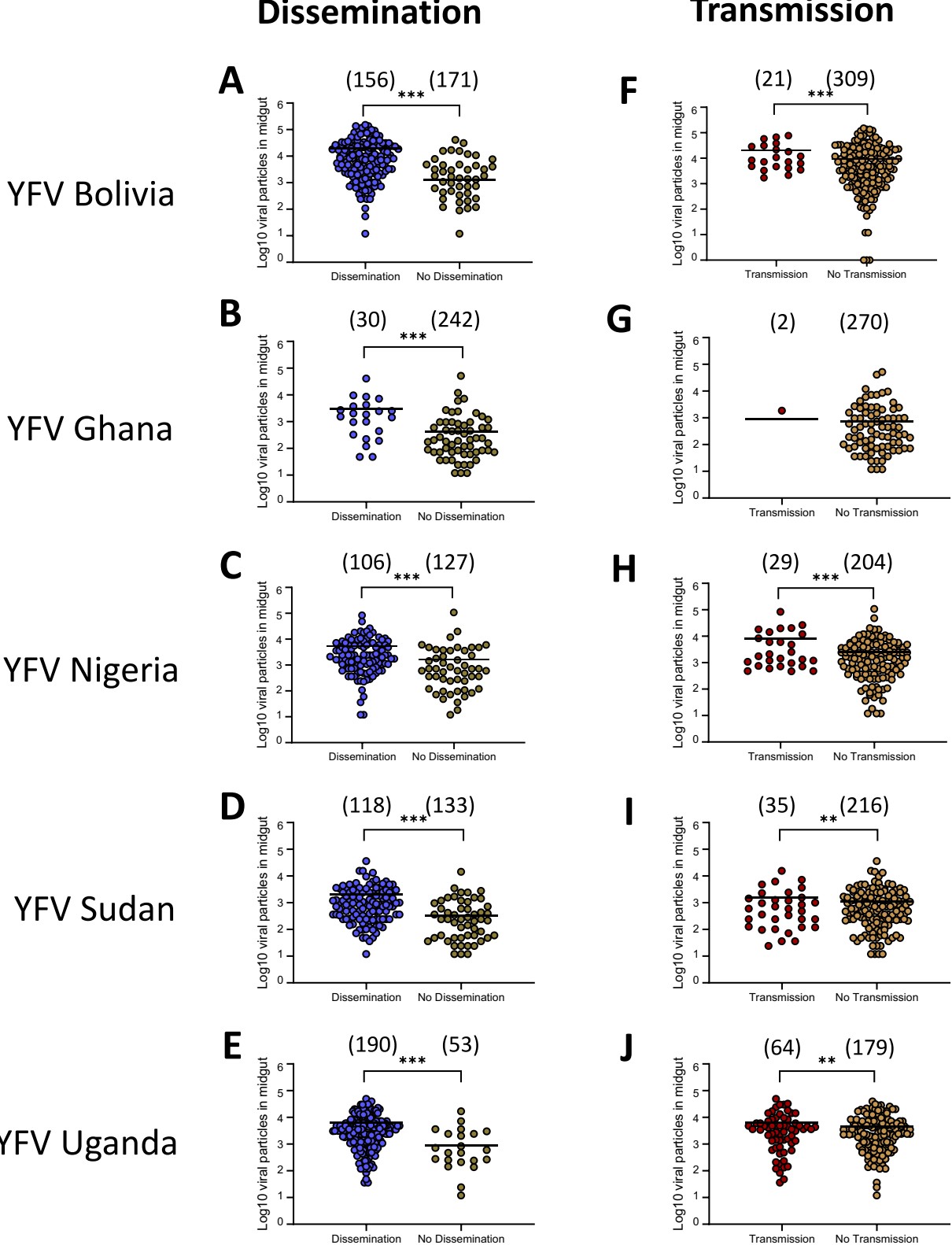

*thuringiensis var israelensis* (*Bti*, Vectobac®) is used as a larvicide since 2010 and deltamethrin (K-Othrine 15/5®) as an adulticide since 1986[46]. *Ae. aegypti* populations from Martinique have unfortunately become resistant to deltamethrin[45]. Alarmingly, it has been reported that insecticide resistance may increase pathogen transmission[47]. Therefore, developing tools for early detection of human cases, quantifying the risk of spread by viremic travelers from YF-endemic regions, and being prepared for a rapid response by mass vaccination are the only means left at our disposal. Moreover, limiting spread to neighboring countries calls for a strong cross-border cooperation and information sharing. Therefore, genomic, and phenotypic surveillance of YFV strains should be augmented, and phylogenetic tools be utilized more frequently to predict outbreaks and develop appropriate countermeasures.

**Fig. 5 | Number of virus particles (in Log$_{10}$) detected in the midgut of nine *Aedes aegypti* populations from Martinique according to mosquito status: with/ without dissemination and with/without transmission.** Mosquitoes were examined 21 days after an infectious meal containing one YFV strain (Bolivia, Ghana, Nigeria, Sudan, and Uganda) provided at 10$^7$ FFU/mL. Legs and wings of individual mosquitoes were removed, and the proboscis was inserted into a pipette tip to collect saliva. Then, mosquitoes were dissected to isolate the midgut from the carcass. The number of virus particles in midgut, carcass, and saliva were estimated by titration. **A**–**E** The number of viral particles in individual midguts (scatter plot) and mean (bar) were estimated for mosquitoes able to disseminate the virus (detection of virus in carcass) *versus* mosquitoes unable to disseminate the virus (no detection of virus in carcass); ***0.0001 ≤ $p$ < 0.001 ($p$ = 0.0001 for **A**–**E**) by Kruskall-Wallis non-parametric test (one-sided). **F**–**J** The number of viral particles in individual midguts and mean (bar) were estimated for mosquitoes able to transmit the virus (detection of virus in saliva) *versus* mosquitoes unable to transmit the virus (no detection of virus in saliva); **0.001 ≤ $p$ < 0.01 ($p$ = 0.0044 for **I**, $p$ = 0.0077 for **J**), ***$p$ < 0.001 ($p$ = 0.0001 for **F**, **H**) by Kruskall-Wallis non-parametric test (one-sided). In brackets are the numbers of mosquitoes tested. Source data are provided as a Source Data file.

## Methods

### Ethic statements
Animals were housed in the Institut Pasteur animal facilities accredited by the French Ministry of Agriculture for performing experiments on live rodents. Work on animals was performed in compliance with French and European regulations on care and protection of laboratory animals (EC Directive 2010/63, French Law 2013-118, February 6th, 2013). All experiments were approved by the Ethics Committee #89 and registered under the reference APAFIS ("Autorisation de Projet utilisant des Animaux à des Fins Scientifiques") #6573-201606l412077987 v2.

### Mosquito populations
Fifteen *Ae. aegypti* populations were used: nine from Martinique, two from Haiti, one from Guadeloupe, one from Cuba, one from Florida (USA), and one from Colombia (Table S1). Eggs laid by females on blotting papers were sent to the Institut Pasteur in Paris. Once received, eggs were immersed in dechlorinated water at 25 °C ± 1 °C for hatching. After 24 h, larvae were distributed in batches of 200 per pan containing one liter of dechlorinated water supplemented with a yeast tablet renewed every 2 days and maintained at 25 ± 1 °C. Pupae were collected in plastic pots placed in cages where adults emerged. Adults were fed with a 10% sucrose solution and kept at 28 ± 1 °C with a 12 L:12D cycle and 80% relative humidity. To amplify the population, females were blood-fed three times a week on anaesthetized female mice (strain IOPS OF1, Charles River laboratories, France). Generations F2 to F6 were used for experimental infections (Table S1).

### Virus strains
Five YFV strains (Bolivia, Ghana, Nigeria, Sudan and Uganda) were obtained via the EVAg consortium (https://www.european-virus-archive.com/). The viral stocks were produced on *Ae. albopictus* C6/36 cells (ATCC®, Virginia, USA) and stored at −80 °C until use. Viral titers are expressed as FFU/mL (focus-forming units/mL). The Bolivia strain was isolated in 1999 and belonged to the American II genotype (GenBank accession number: MF004382), the Nigeria strain in 1970 (West African I genotype, AF369681), the Ghana strain in 1927 (West African II genotype, MF405338), the Uganda strain in 1948 (East African genotype; sequence from https://www.european-virus-archive.com/virus/yellow-fever-virus-strain-uveyfv1948ugmr896-tvp3236), and the Sudan strain in 1940 (East/Central African genotype, MF004383) (Fig. S1). All YFV strains were isolated from patients except the Uganda strain, which was isolated from mosquitoes.

### Mosquito infections and processing
Ten-to-15-day-old females were cold anaesthetized, placed in plastic boxes and exposed to an infectious blood meal. The infectious blood meal containing 1.4 mL of washed rabbit erythrocytes, 700 μL of viral suspension, and ATP at 1 mM as a phagostimulant, was placed in a capsule covered with a pork intestine as membrane and maintained at 37 °C. The virus titer of the blood meal was 10$^7$ FFU/mL (Fig. S3). After 30 min of feeding, engorged mosquitoes were transferred into cardboard boxes and supplied with 10% sucrose under controlled conditions (28 ± 1 °C, relative humidity of 80%, 12 L:12D cycle). Mosquitoes were examined at 14 and 21 days post-infection (dpi). Briefly, female mosquitoes were cold anesthetized, legs and wings were removed and the proboscis was inserted into a pipette tip containing 5 μL of foetal bovine serum (FBS; Eurobio Scientific, Les Ulis, France)[48]. After 30 min, the tip content was retrieved in 45 μL of L-15 medium (ThermoFisher Scientific, Massachusetts, USA). Then, mosquitoes were dissected to isolate the midgut from the carcass. Midguts and carcasses were individually homogenized in 300 μL of L-15 supplemented with 3% FBS and centrifuged at 11,000 x *g* for 5 min. The presence and number of virus particles in each sample (midgut, carcass, and saliva) were estimated by titration.

As recently field-collected mosquitoes fed poorly on an artificial feeding system in BSL-3, the number of engorged mosquitoes was usually low, limiting us to run one single experiment for each combination YFV – mosquito population - dpi.

### Virus titration
The 96-well plates were seeded with C6/36 cells in L-15 supplemented with 10% FBS and incubated for 48 h at 28 °C. Wells were then inoculated with serial 10-fold dilutions of samples in L-15 supplemented with 2% FBS and incubated for 1 h at 28 °C. After incubation, the inoculum was removed and 150 μL of a mixture (1:3) of 4% carboxymethylcellulose (CMC) (Sigma-Aldrich, Missouri, USA) and L-15 supplemented with 8% FBS were added. To avoid contamination, an antifungal-antibacterial mixture (penicillin, streptomycin, and amphotericin) (Life Technologies, California, USA) was added for a final concentration of 1.5X. After incubation for 5 days at 28 °C, cells were fixed with a formaldehyde solution (3.4%) (Sigma Aldrich) for 20 min at room temperature. After 3 washes with 1X PBS and incubation for 15 min in 0.5% TritonX-100™ solution (Sigma Aldrich), cells were labeled with an anti-YFV mouse monoclonal antibody (clone: 0G5) (#NB100-64510, Life Technologies) diluted 1:400 in 1X PBS containing 1% bovine serum albumin (BSA) (Interchim, Montluçon, France) for 1 h at 37 °C. Then, cells were rinsed 3 times with 1X PBS followed by the addition of an Alexa fluor 488-conjugated goat anti-mouse secondary antibody, diluted 1:500 (Thermofisher Scientific). After a 30 min incubation at 37 °C, cells were rinsed 3 times with 1X PBS. Titers were determined by visualizing virus foci under a fluorescence microscope.

### Vector competence indices
Vector competence is determined by measuring three parameters, each corresponding to the spreading of viral particles into successive compartments of the mosquito: (i) the infection rate (IR) corresponding to the proportion of mosquitoes with an infected midgut among mosquitoes exposed to the blood meal, (ii) the stepwise dissemination rate (SDR) referring to the proportion of mosquitoes with an infected carcass (capable of disseminating the virus beyond the midgut and infecting secondary organs or tissues) among mosquitoes with an infected midgut, and (iii) the stepwise transmission rate (STR) corresponding to the proportion of mosquitoes with infectious saliva among mosquitoes with an infected carcass.

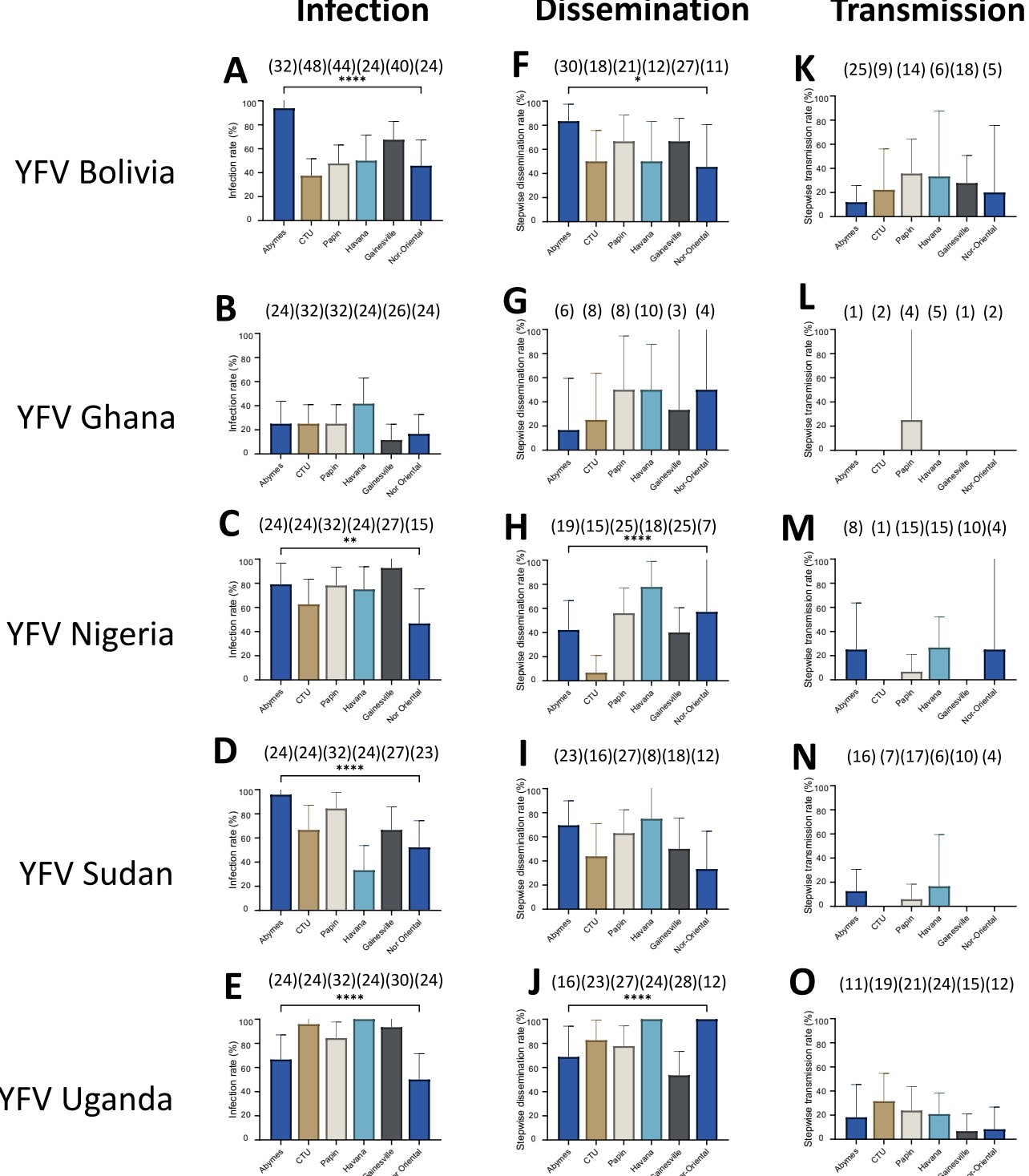

**Fig. 6 | Infection rates, stepwise dissemination rates, and stepwise transmission rates of six *Aedes aegypti* populations from the Caribbean-Americas region examined 21 days after exposure to an infectious blood meal containing one YFV strain (Bolivia, Ghana, Nigeria, Sudan or Uganda).** After exposure to an infectious blood meal provided at a titer of 10⁷ FFU/mL, engorged mosquitoes were kept for 21 days in controlled conditions until examination. Legs and wings of individual mosquitoes were removed, and the proboscis was inserted into a pipette tip to collect saliva. Then, mosquitoes were dissected to isolate the midgut from the carcass. The number of virus particles in midgut, carcass, and saliva were estimated by titration. **A–E** The infection rate was defined as the proportion of mosquitoes with an infected midgut among the total number of mosquitoes exposed to the blood meal (Table S4). **F–J** The stepwise dissemination rate was defined as the proportion of mosquitoes with an infected carcass among mosquitoes with an infected midgut (Table S4). **K–O** The stepwise transmission rate was defined as the proportion of mosquitoes with infectious saliva among mosquitoes with an infected carcass (Table S4). The error bars correspond to the confidence intervals (95%) for IR (**A–E**), SDR (**F–J**), and STR (**K–O**). *$0.01 \leq p < 0.05$, **$0.001 \leq p < 0.01$, ****$p < 0.0001$ by Fisher's exact test (two-sided). In brackets are the numbers of mosquitoes tested. Source data are provided as a Source Data file.

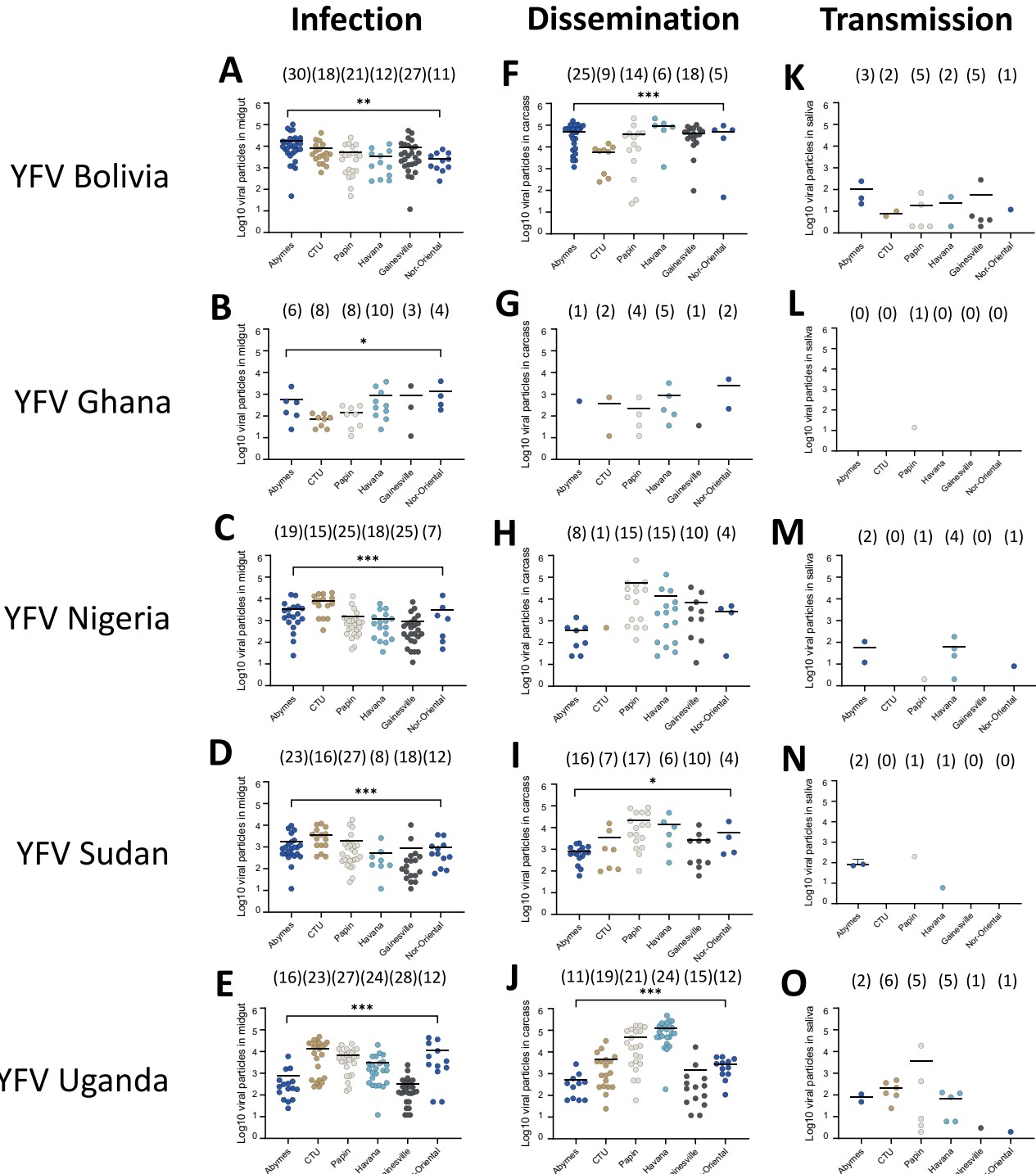

**Fig. 7 | Viral loads in midguts, carcasses and saliva of six *Aedes aegypti* populations from the Caribbean-Americas region examined 21 days after exposure to an infectious blood meal containing one YFV strain (Bolivia, Ghana, Nigeria, Sudan or Uganda).** After exposure to an infectious blood meal provided at a titer of $10^7$ FFU/mL, engorged mosquitoes were kept for 21 days in controlled conditions until examination. Legs and wings of individual mosquitoes were removed, and the proboscis was inserted into a pipette tip to collect saliva. Then, mosquitoes were dissected to isolate the midgut from the carcass. The number of virus particles in midgut, carcass, and saliva were estimated by titration. **A–E** The number of viral particles in individual mosquito midguts (scatter plot) and mean (bar) (Table S4). **F–J** The number of viral particles in individual mosquito carcasses (scatter plot) and mean (bar) (Table S4). **K–O** The number of viral particles in individual mosquito saliva (scatter plot) and mean (bar) (Table S4). *$0.01 \leq p < 0.05$ ($p = 0.0267$ for **B**, $p = 0.0143$ for **I**), **$0.001 \leq p < 0.01$ ($p = 0.0015$ for **A**), ***$0.0001 \leq p < 0.001$ ($p = 0.0001$ for **C–E, J**) by Kruskall-Wallis non-parametric test (one-sided). Bars indicate the mean. In brackets are the numbers of mosquitoes tested. Source data are provided as a Source Data file.

SDR measures the efficiency of the midgut as a barrier to the dissemination of the virus from the midgut into the hemocoel; the higher SDR is, the less the midgut acts as a barrier to the dissemination of the virus. In addition to SDR, STR measures the efficiency of the salivary glands as a barrier to the excretion of the virus in the saliva. Like SDR, the higher STR is, the salivary glands play a lesser role of barrier to the transmission of the virus. IR, SDR, and STR are three parameters that should be considered together[49]. Another parameter, the cumulative transmission rate (CTR), which refers to the proportion of mosquitoes with infectious saliva among mosquitoes tested, measures the transmission efficiency without considering dissemination and infection[49,50]. The CTR is the product of IR, SDR and STR.

## Modeling of vector competence

A natural quantity of interest for studying the capacity of a vector to transmit back YFV to human hosts is the probability that a mosquito (from a population denoted $p$) successfully acquires a given YFV genotype (denoted $g$) upon an infectious blood meal, and that viral particles successfully pass the successive barriers towards the salivary glands. We estimated the CTR in a Bayesian framework, so that uncertainty from the various predictors in the model would be propagated into its posterior distributions. To achieve successful ability to transmit, three successive steps must be met: *i)* successful infection during a blood meal; *ii)* passage of the midgut barrier; and *iii)* passage of the salivary gland barrier. In controlled infection experiments, the viral load in blood meals is presumed fixed and therefore no density-dependent effect can be accounted for. Steps *ii)* and *iii)* however, may be modulated by the viral load in the previous compartment. Denoting $Y_{inf}$, $Y_{dis}$, and $Y_{tra}$ the binomial variables coding for a success in infection, dissemination and transmission (the latter relating directly to CTR), and since passage all the way towards salivary glands implicates that the previous two binomial outcomes were a success, we posit the following model:

$$P\left(Y_{tra}=1\right)=P\left(Y_{inf}=1\right)*P\left(Y_{dis}=1|Y_{inf}=1\right)*P\left(Y_{tra}=1|Y_{inf}=1,Y_{dis}=1\right)$$
$$(1)$$

The corresponding log-likelihood function $\ell_\theta$ is split across three terms, each related to a binomial distribution resulting from its own set of parameters ($\theta_{inf}$, $\theta_{dis}$, $\theta_{tra}$):

$$\ell_\theta\left(Y_{tra}\right)=\ell_{\theta_{inf}}\left(Y_{inf}\right)+\ell_{\theta_{dis}}\left(Y_{dis}\right)+\ell_{\theta_{tra}}\left(Y_{tra}\right)$$
$$(2)$$

Binomial models for all three Y outcomes allowed the probability of success to vary according to days post-infection (coded as categorical), mosquito population ($p$), YFV genotype ($g$), and included an interaction term between these two predictors ($gp$). Dissemination and transmission were furthermore allowed to vary according to the log10-transformed viral load ($v$). The full set of parameters was therefore:

$$\begin{cases}\theta_{inf}=(\mu_{inf},\beta_{dpi_1},\beta_{p_1},\beta_{g_1},\beta_{gp_1})\\\theta_{dis}=(\mu_{dis},\beta_{dpi_2},\beta_{p_2},\beta_{g_2},\beta_{gp_2},\beta_{v_2})\\\theta_{tra}=(\mu_{tra},\beta_{dpi_3},\beta_{p_3},\beta_{g_3},\beta_{gp_3},\beta_{v_3})\end{cases}$$

The model was developed in Stan using the rstan package (v. 2.21.8) in R (v. 4.3.0). Four independent chains of 5,000 iterations were run, the second half serving for sampling. Convergence was assessed using the Gelman-Rubin R-hat statistic[51].

Weakly informative priors were used for the logistic regression coefficients when building the joint Bayesian model, as recommended by Gelman et al.[52]. These consisted in Student distributions with

means = 0 and standard deviation = 2.5, thus covering a wide range of odds-ratios on a logit scale. Model intercepts used a Normal prior with mean = 0 and standard deviation = 10.

## Statistical analyses

Proportions were compared using Fisher's exact test and viral loads using Kruskall-Wallis non-parametric tests. Analyses were conducted using the STATA software (StataCorp LP, Texas, USA). $P$-values $< 0.05$ were considered statistically significant.

## Reporting summary

Further information on research design is available in the Nature Portfolio Reporting Summary linked to this article.

## Data availability

The data that support the findings of this study are available in the supplementary tables. Source data are provided with this paper.

## Code availability

All analyses and model results presented in this manuscript and associated code will are available at the following GitLab repository: https://gitlab.pasteur.fr/tobadia/20230619-yfv-vector-competence/. The analyses and model results presented in this paper correspond to commit 6ed3b59af545085d42f39544d61f7eb6b9422d95.

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

## Acknowledgements

Authors thank Drs. Heather Coatsworth and Caroline J. Stephenson for collecting mosquitoes in Gainesville, Florida, as well as Malika Hocine for her help in rearing mosquitoes and Patricia Smiley for proofreading. We also thank Prof. André Cabié (CHU of Martinique) for his support. This work was supported by the Laboratoire d'Excellence "Integrative Biology of Emerging Infectious Diseases" (grant n°ANR-10-LABX-62-IBEID), and the "Agence Régionale de Santé" (ARS) of Martinique. GG received a doctoral fellowship from the Territorial Collectivity of Martinique. The funders had no role in study design, data collection and interpretation, or decision to submit the work for publication.

## Author contributions

G.G., L.M., C.B., M.V.I. and M.V.A. did the investigation. T.O. and C.D. performed data analysis and interpretation. T.O., M.V.A., C.B., M.V.I., R.R.D., A.Y., A.V.R., G.G.B., R.M.G.R, X.D.L. participated in manuscript revision. R.R.D., A.Y., A.V.R., G.G.B., R.M.G.R, F.S. and M.E. collected mosquito populations. N.D.P., A.B., J.S.R. and X.D.L. contributed to supervision. ABF intervened in supervision, conception and design, funding acquisition, and writing original draft. All authors have read and agreed with the published version of the manuscript.

## Competing interests

The authors declare no competing interests.
