## [Peer Review File · Nature Communications]

Using vector competence data to characterize Yellow fever transmission in the CaribbeanREVIEWER COMMENTS

Reviewer #1 (Remarks to the Author):

This is a comprehensive study determining variation in vector competence of populations of *Aedes aegypti* in the Caribbean for Yellow Fever virus (YFV). YFV has been a major scourge of humanity. A highly effective vaccine is available but the disease has reemerged on multiple occasions due to reasons including, the persistence of the virus in animal reservoirs, failures of vaccination programs and the high prevalence of the primary vector *Aedes aegypti* in endemic regions. This was a highly comprehensive study that initially compared 45 *Aedes aegypti* / virus strain pairings (9 populations vs 5 virus strains) across the Caribbean, followed by a comparison of 30 population-virus combinations within Martinique. This is one of the largest comparisons of mosquito susceptibility to an arbovirus. Distinct geographic and mosquito-virus strain differences in susceptibility were demonstrated. The study tested mosquito colonies with low generation numbers, maintaining their representativeness of field populations. The vector competence experiments were well executed and the manuscript well written.

Main comments

L654-696 and Figs 2-7. Please describe key aspects of the figures in the legends; including; what do the bars represent in A, B, C, D and E? What does colouration of the bars mean (some bars are coloured and other uncoloured)? What do the error bars represent (i.e. SD, SE, range) and how many replicates were performed? What do the points signify in F, G, H, I and J? Are these titres for individual mosquitoes? What do the lines represent (means/medians)? Please state the tests applied in the figure legends.

The figures are scientifically sound and presentable (with the amendments described above). However, I found it difficult to try to identify geospatial trends in the infection prevalence and intensity data between the different locations. I found myself constantly referring back to Fig 1 to try to identify geospatial trends. I believe this would also be of interest to other readers. There may be a better way to group the regions on the x-axis (currently grouped in alphabetical order). The regions could possibly be ordered on the x-axis according to their latitude, or according to the highest to lowest YFV prevalence, although the pattern changes for different YFV strains and sample types and would mean the order changes from graph to graph. The infection rates could possibly be depicted in pie charts superimposed onto the position of the location on a map; though again the patterns change between YFV strains and sample types. Consider the segmented pie chart approach used to depict virus susceptibility to different strains superimposed on maps in Aubrey et al. 2020 Science PMID:33214283. I don't have an answer and this is a suggestion only. L459-464. The authors briefly mention the study demonstrating *Aedes aegypti* are generically structured in the region (ref 48), the paper demonstrating geographic variation in Zika virus susceptibility among *Aedes aegypti* (ref 18) and the broad differences in susceptibility between the populations they tested. Are there any trends in the susceptibility data from their studies to differences in *Aedes aegypti* genotypes or estimated historic movements of the species in the region? Please discuss.

Minor

L95. Change "result" to "resulting"

L114. Change "is thought" to "are thought" and "virusxvector" to "virus × vector"

L186 Please state the name of the antibacterial-antifungal mixture.

L199. I don't believe "diffusion" is the right term to describe the spread of virus through the mosquitoes. Diffusion suggests a passive process of an inanimate compound, rather than active process of replication and spread of new virions.

L202 I am familiar with the indices used but this is the first time I have encountered them being called "stepwise" dissemination and transmission measures. Please cite the manuscript providing the original usage of the term if used previously.

L222 confirm the "joint probability of transmission" is different to the "cumulative transmission rate"

L310 and 314. Change "can" to "could"

L319. It is not clear what "(respectively carcass)" means. Do the authors mean "(and subsequently in the carcass)"? However, the structure of the sentence is clumsy.

L320. Same comment as above for "(resp. transmission)".

L321. Consider using "without" instead of "after dropping"

L419-420. Statement "A limitation of our approach resides in the observed data used for model fitting, in particular related to viral loads, the only continuous predictor." The sentence does not state what the problem is with the observed data.

Currently, the strategy for grouping panels for figures 6 and 7 is different to figures 2-4 (grouping infection rates together (Fig 6 and 7) vs grouping tissues together (Figs 2-4). I suggest a consistent approach is used.

Reviewer #2 (Remarks to the Author):

This work attempts to estimate the risk of yellow fever (YF) in the Caribbean using vector competence data. The authors investigate the variation in mosquito's physiological ability to support parasite development, formally known as vector competence, by exposing mosquitoes from different geographic regions to infectious blood meals containing one among five YFV genotypes. Detection of virus in mosquito saliva is used as a proxy for competence to infect additional hosts.

The results of this work could potentially help us answer current puzzles regarding YF transmission, such as the presence of the sylvatic pattern of transmission in South and Central American countries as a potential source of spillover mechanisms leading to the disease's urban pattern of transmission seen in the past. Could this puzzle be explained by the lack of competence of current urban populations of mosquitoes to transmit current strains of virus in urban settings? Are other pathogens also transmitted by the same mosquito species vector, outcompeting the YFV?

In my understanding, this work does not provide clear clues to answer the relevant questions regarding the re-emergence of urban YF transmission originating from local sylvatic cycles or disease importation from other regions due to the following limitations:

i) Vector competence is a necessary but not sufficient condition for transmission. The ability of a vector to transmit a pathogen is classically described as the vectorial capacity, formally defined as the maximal potential force of pathogen transmission by a local vector population. Vectorial capacity is determined by a combination of four attributes: physiological ability to support pathogen development (vector competence), average daily size of the host-seeking population, average adult longevity of female mosquitoes, and the proportion of the mosquito population that feeds on blood. Vector competence is a necessary condition for transmission, but it does not, per se, allow us to estimate the RISK of re-emergence (urbanization) or introduction (spillover). Therefore, the concept of "risk estimation" as it appears in the title seems to overstate the implications of the work.

ii) The study does not properly handle the role of evolutionary changes in vector competence. Why some members of the same vector species or genus transmit a pathogen while others do not, or are less efficient, is no doubt of intense interest to vector biologists. As a first approximation, we cannot exclude the role of both rapid and gradual evolutionary changes relevant to vector ability. Given the high evolvability of the host-pathogen relationship, assessment of vector competence requires that mosquito and virus populations would match in time to better represent current genetic standing variation ("specific pairing of vector and virus genotype"). This is not the case in this work given the isolation dates of the virus strains (Bolivia-1999, Nigeria-1970, Ghana-1927, Uganda-1948, Sudan-1940) used to challenge the vector populations.

iii) The study does not mimic natural conditions of virus transmission. Since the virus concentration of the infectious blood meals determine the experimental outcome observed, presence of virus in the saliva as a proxy for transmission, a translation of the results to the field conditions would require that the experiments were conducted mimicking the distribution of actual level of mammalian host viremia. Infectious blood meals containing higher virus concentrations

than found in natural conditions in the mammalian host would bias the final assessment of competence overestimating it.

In conclusion, while this work provides valuable insights into the variation in vector competence among mosquito populations from different geographic regions, it does not provide clear clues to answer the relevant questions regarding the re-emergence of YF transmission.

Additional comments:

Line 96: I could not follow the sentence starting in line 96.

Line 238: The equation looks like the log-posterior to me and not the log-likelihood. This equation does not add much to the description of the model and can be removed.

Line 245: The equations describe the labels assigned to the relevant parameters. They do not provide much information about the functional structure of the model.

Line 270: I had difficulty identifying the null hypothesis that is being tested by the Fisher exact test.

Line 274: The same problem as above now for the Kruskal-Wallis test.

Answers to Reviewer #1

Point #1

L654-696 and Figs 2-7.

Please describe key aspects of the figures in the legends; including; what do the bars represent in A, B, C, D and E?

What does colouration of the bars mean (some bars are coloured and other uncoloured)?

What do the error bars represent (i.e. SD, SE, range) and how many replicates were performed?

What do the points signify in F, G, H, I and J?

Are these titres for individual mosquitoes?

What do the lines represent (means/medians)?

Please state the tests applied in the figure legends.

We have organized the X axis with the names of mosquito populations in alphabetic order which we believe is the easiest way to present the results considering the small size of the island. Bars colors come from a format proposed by the software with no precise meaning. We have added the missing details in the legends of the figures (2-7): confidence intervals, means, statistical tests...

Point #2

The figures are scientifically sound and presentable (with the amendments described above). However, I found it difficult to try to identify geospatial trends in the infection prevalence and intensity data between the different locations. I found myself constantly referring back to Fig 1 to try to identify geospatial trends. I believe this would also be of interest to other readers. There may be a better way to group the regions on the x-axis (currently grouped in alphabetical order). The regions could possibly be ordered on the x-axis according to their latitude, or according to the highest to lowest YFV prevalence, although the pattern changes for different YFV strains and sample types and would mean the order changes from graph to graph. The infection rates could possibly be depicted in pie charts superimposed onto the position of the location on a map; though again the patterns change between YFV strains and sample types. Consider the segmented pie chart approach used to depict virus susceptibility to different strains superimposed on maps in Aubrey et al. 2020 Science PMID:33214283. I don't have an answer and this is a suggestion only.

In Aubry's paper (doi: 10.1126/science.abd3663), the authors only examined the infection rate of mosquitoes. Seven days after the infectious blood meal, mosquitoes are processed to extract viral RNA for RT-PCR reaction. In our case, we have calculated the infection, stepwise dissemination and stepwise transmission rates to estimate the importance of the midgut and the salivary glands in virus dissemination and transmission, respectively. I believe it will be tough to present infection, dissemination and transmission in a same figure.

Point #3

L459-464. The authors briefly mention the study demonstrating *Aedes aegypti* are generically structured in the region (ref 48), the paper demonstrating geographic variation in Zika virus susceptibility among *Aedes aegypti* (ref 18) and the broad differences in susceptibility between the populations they tested. Are there any trends in the susceptibility data from their studies to differences in *Aedes aegypti* genotypes or estimated historic movements of the species in the region? Please discuss.

Aedes aegypti is native to tropical forests in Sub Saharan Africa. The domestication of *Ae. aegypti* was the result of the Sahara Desert expansion (4000–6000 years ago). *Ae. aegypti* evolved to breed in man-made containers and to take blood meals from humans. The domestic form was later introduced into the New World during the slave trade beginning in the 1500s and arrived in Asia in the 1890s. The species persisted in the Mediterranean Basin until about 1950. The species invaded the Pacific region after World War II.

Based on Gloria-Soria's paper (doi: 10.1111/mec.13866.), *Aedes aegypti* populations outside Africa are subdivided into 3 major groups: North America, South America and Asia-Pacific. We

have shown in a previous paper that 26 *Aedes aegypti* populations collected in Martinique in 2001-2002 were genetically differentiated (Yebakima et al. 2004; doi: 10.1111/j.1365-3156.2004.01241.x.).

We have added at:

- L453-L456: "In Martinique, *Ae. aegypti* populations are expected to derive from mosquitoes from North America and South America. These populations were shown to be genetically differentiated most likely as a result of recurrent insecticide treatments in response to dengue outbreaks ⁵⁰."
- L621-623: the reference 50- Yebakima A, Charles C, Mousson L, Vazeille M, Yp-Tcha MM, Failloux AB. Genetic heterogeneity of the dengue vector *Aedes aegypti* in Martinique. *Trop Med Int Health*. 2004 May;9(5):582-7. doi: 10.1111/j.1365-3156.2004.01241.x. PMID: 15117302.

Point #4

L95. Change "result" to "resulting"

We have modified.

Point #5

L114. Change "is thought" to "are thought" and "virusxvector" to "virus × vector"

We have changed.

Point #6

L186 Please state the name of the antibacterial-antifungal mixture.

We have added "(penicillin, streptomycin, and amphotericin)".

This solution contains 10,000 units / ml penicillin (Penicillin G Sodium Salt), 10,000 µg / ml streptomycin (Streptomycin sulfate) and 25 µg / ml Gibco B amphotericin.

Point #7

L199. I don't believe "diffusion" is the right term to describe the spread of virus through the mosquitoes. Diffusion suggests a passive process of an inanimate compound, rather than active process of replication and spread of new virions.

We suggest using "spreading".

Point #8

L202 I am familiar with the indices used but this is the first time I have encountered them being called "stepwise" dissemination and transmission measures. Please cite the manuscript providing the original usage of the term if used previously.

In addition to the reference 20 (Bisia, M. *et al.* Secondary vectors of Zika Virus, a systematic review of laboratory vector competence studies. *PLoS Negl Trop Dis* **17**, e0011591, doi:10.1371/journal.pntd.0011591 (2023)), we have added another reference:

Reference 21 – Wu VY, Chen B, Christofferson R, Ebel G, Fagre AC, Gallichotte EN, Sweeny AR, Carlson CJ, Ryan SJ. A minimum data standard for vector competence experiments. *Sci Data*. 2022 Oct 19;9(1):634. Doi: 10.1038/s41597-022-01741-4. PMID: 36261651; PMCID: PMC9582208.

Point #9

L222 confirm the 'joint probability of transmission' is different to the "cumulative transmission rate".

That "joint probability of transmission" actually corresponds to the CTR (=IR * SDR * STR, as was mentioned at L218). The reviewer was right in pointing out non-harmonized notations and we now refer to as CTR all throughout the text, for consistency with the currently accepted/suggested terminology from Wu *et al.* and Bisia *et al.* Minor edit made at L218 for clarity.

We had initially used the term “joint probability” to denote a quantity that was informed by more than observed raw proportions (IR, SDR, STR) as these are also modelled, but this did not make our message clearer.

Point #10

L310 and 314. Change “can” to “could”

We have changed.

Point #11

L319. It is not clear what “(respectively carcass)” means. Do the authors mean “(and subsequently in the carcass)”? However, the structure of the sentence is clumsy.

We have rephrased that sentence for clarity (L310-311): “Logistic regression revealed the viral load in midguts to be an important predictor in explaining dissemination, and likewise the viral load in the carcass in explaining transmission.”

Point #12

L320. Same comment as above for “(resp. transmission)”.

See Point #11.

Point #13

L321. Consider using “without” instead of “after dropping”

We have changed.

Point #14

L419-420. Statement “A limitation of our approach resides in the observed data used for model fitting, in particular related to viral loads, the only continuous predictor.” The sentence does not state what the problem is with the observed data.

We have modified that sentence to better describe the limitation (L405-409): “An implicit limitation of our model resides in the representativity of viral loads (the only continuous predictors) used for model fitting: our model makes the underlying assumption that the variability of such titers across YFV genotypes and mosquito heterogeneity is well captured from experimental conditions, so that posterior distributions can be used when sampling outcomes of interest.”

Since viral loads in midguts and carcass is used for model fitting (to estimate YFV/mosquito specific Betas), and subsequently for sampling large numbers of individual-based simulations for infection/dissemination/transmission, our method makes the underlying assumption that the distributions of titers, pooled by YFV genotype and mosquito population, is representative enough to allow for generalization of the results.

Point #15

Currently, the strategy for grouping panels for figures 6 and 7 is different to figures 2-4 (grouping infection rates together (Fig 6 and 7) vs grouping tissues together (Figs 2-4). I suggest a consistent approach is used.

The figures 2-4 present 9 populations while the figures 6-7 present 6. We can split the figures 6-7 into 3 figures like the figures 2-4 leading to have an additional figure. We chose to limit the number of figures where possible.

Answers to Reviewer #2

Point #1

Vector competence is a necessary but not sufficient condition for transmission. The ability of a vector to transmit a pathogen is classically described as the vectorial capacity, formally defined as the maximal potential force of pathogen transmission by a local vector population. Vectorial capacity is determined by a combination of four attributes: physiological ability to support pathogen development (vector competence), average daily size of the host-seeking population, average adult longevity of female mosquitoes, and the proportion of the mosquito population that feeds on blood. Vector competence is a necessary condition for transmission, but it does not, per se, allow us to estimate the RISK of re-emergence (urbanization) or introduction (spillover). Therefore, the concept of "risk estimation" as it appears in the title seems to overstate the implications of the work.

We agree with the reviewer's comment. The vector competence is a pre-requisite for a transmission to occur but other factors as mentioned, are included to assess the vectorial capacity which in short, refers to the role of a mosquito species as vector of a given pathogen on the field. If the mosquito is not competent, no transmission occurs even if the mosquito is present in high densities, bites humans, and survive long enough to transmit. We suggest to add transmission in the title: "Estimating the risk of Yellow fever **transmission** in the Caribbean using vector competence data.

Point #2

The study does not properly handle the role of evolutionary changes in vector competence. Why some members of the same vector species or genus transmit a pathogen while others do not, or are less efficient, is no doubt of intense interest to vector biologists. As a first approximation, we cannot exclude the role of both rapid and gradual evolutionary changes relevant to vector ability. Given the high evolvability of the host-pathogen relationship, assessment of vector competence requires that mosquito and virus populations would match in time to better represent current genetic standing variation ("specific pairing of vector and virus genotype"). This is not the case in this work given the isolation dates of the virus strains (Bolivia-1999, Nigeria-1970, Ghana-1927, Uganda-1948, Sudan-1940) used to challenge the vector populations.

Unfortunately, we do not have access to recently YFV isolates which is a limitation of our study. We made the choice to select five isolates which illustrate 5 genotypes among the 7 existing ones. The objective is to measure the variation of vector competence using 5 YFV genotypes. We are aware that the reviewer's comment opens a very interesting research topic which we have partially explored by adapting experimentally YFV to a new invasive vector, *Ae. albopictus*. We did 10 *in vivo* passages of YFV in *Ae. albopictus*, and detected several genetic changes in the viral genome including substitutions in the NS1 gene which plays a role in eliciting the host immune response. These experiments show that the viral genome can become more adapted to a new vector with an increase of virus titers in the mosquito saliva which could facilitate urban outbreaks in regions where it is established.

Amraoui F, Pain A, Piorkowski G, Vazeille M, Couto-Lima D, de Lamballerie X, Lourenço-de-Oliveira R, Failloux AB. Experimental Adaptation of the Yellow Fever Virus to the Mosquito *Aedes albopictus* and Potential risk of urban epidemics in Brazil, South America. *Sci Rep.* 2018 Sep 25;8(1):14337. doi: 10.1038/s41598-018-32198-4. PMID: 30254315; PMCID: PMC6156417.

Point #3

The study does not mimic natural conditions of virus transmission. Since the virus concentration of the infectious blood meals determine the experimental outcome observed, presence of virus in the saliva as a proxy for transmission, a translation of the results to the field conditions would require that the experiments were conducted mimicking the distribution of actual level of mammalian host viremia. Infectious blood meals containing higher virus concentrations than

found in natural conditions in the mammalian host would bias the final assessment of competence overestimating it.

We have chosen the titer of 10^7 FFU/mL for mosquito blood meals because:

- this titer is the typical titer we use in our laboratory, which gives us a level of comparison with other viruses and other virus genotypes
- viral loads detected in patients are usually around 10^6 - 10^8 copies / mL. In Avelino-Silva et al. (2023), the authors found that viremia in patients drop few days after the onset of symptoms: from 10^8 Log₁₀ copies /mL at day 3 to 10^3 at day 10 for survivors to the disease and from 10^{13} at day 3 to 10^4 at day 10 for deceased patients
- We have tested two blood meal titers : 10^5 and 10^7 FFU/mL; no transmission occurs at 10^5 FFU/mL. see supplementary data Figure S3

Point #4

Line 96: I could not follow the sentence starting in line 96.

Sorry for that; we have shortened this sentence which I hope has become clearer.

We have shortened this sentence which I hope has become clearer.

“With >50% of initial YFV exposures resulting in mild or asymptomatic infections, molecular changes in the viral genome may induce additional alterations in YF pathogenesis leading to more severe outbreaks.”

Point #5

Line 238: The equation looks like the log-posterior to me and not the log-likelihood. This equation does not add much to the description of the model and can be removed.

We apologize for the ambiguous notations used in the log-likelihood function. The reviewer is right in pointing out that the conditioning made it look like this was a posterior distribution. We were referring to the likelihood (L) on a log-scale (hence, lowercase). The conditioning bar made it however look like we were referring to a probability distribution. We have now adopted notations $\ell_{\theta}(Y)$ for more clarity (L238-L239).

We also made it clearer (L232-234) that, while integrating, no contribution to the likelihood could be expected when Y_{inf} or Y_{dis} were equal to zero, because their success was a requirement to allow transmission to possibly occur.

Point #6

Line 245: The equations describe the labels assigned to the relevant parameters. They do not provide much information about the functional structure of the model.

While we understand the point raised by the reviewer, we feel these three lines are useful to show how thetas for dissemination and transmission incorporate a distribution for the log₁₀-

transformed viral load (β_v), whereas this is not the case for θ_{inf} . This also echoes the reviewer's point #3, with a single viral concentration used for mosquito exposure to YFV. We agree that removing this set of equations would however reduce length, and propose that if this is still deemed of importance, references to subscripts be removed from the above paragraph (L238-241) so that the reader can still easily understand what parameters each theta consists of.

Point #7

Line 270: I had difficulty identifying the null hypothesis that is being tested by the Fisher exact test.

For more clarity, we have removed all the P-values from the text which do not add provide any additional information for understanding. P-values are indicated in figures.

Point #8

Line 274: The same problem as above now for the Kruskal-Wallis test.

Same response as for #8.

REVIEWERS' COMMENTS

Reviewer #1 (Remarks to the Author):

The authors have addressed my queries and concerns. A more creative arrangement of data in figures could possibly be made (perhaps by illustrators associated with the journal). However, the data are depicted appropriately in the current form.

Minor:

Line 672 "nine *Aedes aegypti* populations from Martinique examined 21 days after exposure to an infectious blood meal containing one YFV strain (Bolivia, Ghana, Nigeria, Sudan, and Uganda)" Suggest replace "and" (in Sudan, and Uganda) to "or" (Sudan or Uganda).

Line 685. As above for changing "Sudan, and Uganda) to (Sudan or Uganda).

Reviewer #2 (Remarks to the Author):

I appreciate the authors' thoughtful and detailed rebuttal to the points raised in my previous review, leading to a productive exchange of ideas. Starting with point #2, where we seem to be in agreement, I suggest adding one or two sentences in the Discussion section addressing the coevolution of virus and the vector host and how the experiments should be interpreted under this scenario.

However, regarding point #1, I believe simply adding the word "transmission" to the title is insufficient to address my concerns. In my interpretation, "risk" is a quantitative concept, typically expressed as a probability within the range [0, 1]. Alternatively, "risk" can be represented as an ordinal variable (e.g., high, medium, low) or a categorical variable (e.g., yes, no). Therefore, upon reading the title, I would expect to find a table with rows and columns detailing locations, virus strains, and distinct vector populations, with each table cell containing a corresponding risk estimate (a number within the range [0, 1]) and its associated uncertainty interval. This is not the case. The word "risk" appears 4 times in the body of the manuscript, none of which in legends of tables or figures.

I suggest adjusting the title to reflect the content of the work more accurately.

Answers to Reviewer #1

Point #1: Line 672 “nine *Aedes aegypti* populations from Martinique examined 21 days after exposure to an infectious blood meal containing one YFV strain (Bolivia, Ghana, Nigeria, Sudan, and Uganda)” Suggest replace “and” (in Sudan, and Uganda) to “or” (Sudan or Uganda).

Line 685. As above for changing “Sudan, and Uganda) to (Sudan or Uganda).

It has been changed in all legends of figures and tables in the main text and the supplementary files.

Answers to Reviewer #2

Point #1: I appreciate the authors' thoughtful and detailed rebuttal to the points raised in my previous review, leading to a productive exchange of ideas. Starting with point #2, where we seem to be in agreement, I suggest adding one or two sentences in the Discussion section addressing the coevolution of virus and the vector host and how the experiments should be interpreted under this scenario.

As requested by the reviewer, we have added few sentences on virus-vector coevolution on lines 435-438: “Vector genetic factors involved in the success of viral transmission result from a long-term virus-vector interactions. Coevolution of vectors and the pathogens they transmit can positively modulate specific gene expression to maintain the vector fitness and secure pathogen transmission.”

Point #2: However, regarding point #1, I believe simply adding the word "transmission" to the title is insufficient to address my concerns. In my interpretation, "risk" is a quantitative concept, typically expressed as a probability within the range [0, 1]. Alternatively, "risk" can be represented as an ordinal variable (e.g., high, medium, low) or a categorical variable (e.g., yes, no). Therefore, upon reading the title, I would expect to find a table with rows and columns detailing locations, virus strains, and distinct vector populations, with each table cell containing a corresponding risk estimate (a number within the range [0, 1]) and its associated uncertainty interval. This is not the case. The word “risk” appears 4 times in the body of the manuscript, none of which in legends of tables or figures. I suggest adjusting the title to reflect the content of the work more accurately.

We understand the reviewer’s comment and have changed the title accordingly: “Using vector competence data to characterize Yellow fever transmission in the Caribbean”.

In the text, we refer to “risk” for different contexts.

- Yellow fever is endemic today in 44 countries in Sub-Saharan Africa and South America where the vector *Ae. aegypti* is present, representing approximately 900 million people at **risk of infection**.
- The French overseas department of French Guiana where vaccination is mandatory, is **at risk of a greater number** of YF imported cases, resulting in local cases, as was in 1998, the first YF outbreak in the country since 1902^{34,35}.
- Therefore, developing tools for early detection of human cases, quantifying **the risk of spread** by viremic travelers from YF-endemic regions, and being prepared for a rapid response by mass vaccination are the only means left at our disposal.